# Determination of Multiple Neurotransmitters through LC-MS/MS to Confirm the Therapeutic Effects of *Althaea rosea* Flower on TTX-Intoxicated Rats

**DOI:** 10.3390/molecules28104158

**Published:** 2023-05-18

**Authors:** Yichen Wang, Renjin Zheng, Pingping Wu, Youjia Wu, Lingyi Huang, Liying Huang

**Affiliations:** 1School of Pharmacy, Fujian Medical University, Fuzhou 350122, China; 2Fujian Provincial Center for Disease Control and Prevention, Physical and Chemical Analysis Department, Fuzhou 350001, China

**Keywords:** tetrodotoxin, LC-MS/MS, *Althaea rosea* flower, neurotransmitters, detoxification

## Abstract

Tetrodotoxin (TTX) inhibits neurotransmission in animals, and there is no specific antidote. In clinical practice in China, *Althaea rosea* (*A. rosea* flower) extract has been used to treat TTX poisoning. In this work, the efficacy of the ethyl acetate fraction extract of *A. rosea* flower in treating TTX poisoning in rats was investigated. A high-performance liquid chromatography–tandem mass spectrometry (LC-MS/MS) method was developed to determine nine neurotransmitters in rat brain tissue, including γ-aminobutyric acid (GABA), dopamine (DA), 5-hydroxytryptamine (5-HT), noradrenaline (NE), 3,4-dihydroxyphenylacetic acid (DOPAC), homovanillic acid (HVA), 5-hydroxyindole-3-acetic acid (5-HIAA), epinephrine (E), and tyramine (Tyn). The detoxifying effect of *A. rosea* flower was verified by comparing the changes in neurotransmitters’ content in brain tissue before and after poisoning in rats. The assay was performed in multiple reaction monitoring mode. The quantification method was performed by plotting an internal-standard working curve with good linearity (R^2^ > 0.9941) and sensitivity. Analyte recoveries were 94.04–107.53% (RSD < 4.21%). Results indicated that the levels of 5-HT, DA, E, and NE in the brains of TTX-intoxicated rats decreased, whereas the levels of GABA, Tyn, and 5-HIAA showed an opposite trend, and HVA and DOPAC were not detected. The levels of all seven neurotransmitters returned to normal after the gavage administration of ethyl acetate extract of *A. rosea* flower to prove that the ethyl acetate extract of *A. rosea* flower had a therapeutic effect on TTX poisoning. The work provided new ideas for studies on TTX detoxification.

## 1. Introduction

Tetrodotoxin (TTX) is a potent neurotoxin found in various creatures in nature, such as puffer fish, some newts, frogs, and a few invertebrate species [1]. Occasional cases of TTX poisoning have been reported in coastal areas of China. TTX is extremely lethal to humans and animals, and the number of TTX-related deaths each year in Japan accounts for more than 70% of all food poisoning deaths during the same period [2]. TTX primarily blocks nerve conduction, causing paralysis of nerve endings and central nerves. The first is sensory nerve paralysis, followed by motor nerve paralysis [3]. It inhibits sodium-channel conductance reversibly and selectively, thereby reducing the effectiveness of transmitter release by preventing nerve conduction [4]. The main symptoms of TTX poisoning from consuming food contaminated with TTX include tingling of the tongue and lips, headache, vomiting, muscle weakness, ataxia, and potentially death from heart failure or respiratory distress [5]. The onset of TTX poisoning is rapid and intense, and no effective treatment options exist, save for supportive care. After TTX poisoning, the quest for therapeutic medications has become a highly pressing clinical issue.

*Althaea rosea* (Linn.) Cavan is a biennial erect herb of the *A. rosea* flower belonging to family *Malvaceae* [6]. According to the National Compilation of Chinese Herbal Medicine, *A. rosea* flower has the effect of detoxification, cough relief, and detoxification of TTX [7]. In China, a decoction of *A. rosea* flower is used for rescue treatment after TTX poisoning with good clinical effects [8,9,10,11]. *A. rosea* flower has been clinically proven to affect the treatment of TTX poisoning, but its pharmacological components and mechanism of action remain unclear.

Neurotransmitters (NTs) and their metabolites directly contribute to the maintenance of a number of physiological processes in the brain, such as the regulation of behavior, mood, and cognition [12]. They are recognized to be important components of the neurological systems of many different creatures [13]. Their main function is to communicate information throughout the brain and the rest of the body [14]. NTs include three main groups: namely, amino acids, biogenic amines, and cholines, as well as their metabolites. The main common amino acid NTs are glycine, glutamic acid, and γ-aminobutyric acid (GABA). Biogenic amine NTs include dopamine (DA), 5-hydroxytryptamine (5-HT), and noradrenaline (NE) and their metabolites 3-methoxytyramine hydrochloride (3-MT), 3,4-dihydroxyphenylacetic acid (DOPAC), homovanillic acid (HVA), and 5-hydroxyindole-3-acetic acid (5-HIAA). The cholinergic NTs are primarily acetylcholine [15,16]. Depending on their effects on postsynaptic neurons, NTs can be subdivided into excitatory and inhibitory. Excitatory transmitters and their receptors can produce excitatory postsynaptic potentials that allow excitation to spread easily and have a pro-convulsant effect, such as the common NE, epinephrine (E), and 5-HT. Lei et al. [17] found a decrease in glutamate, isoleucine, alanine, and α- and β-hydroxybutyrate concentrations after intracortical TTX infusion in rats through a microdialysis technique. Lepiarczyk et al. [18] found that TTX alters the number and distribution of noradrenergic and cholinergic nerve fibers in porcine urinary bladder wall. Thus, the measurement of these components is critical to the toxicological profile of TTX.

Analyzing NTs and their metabolites in biological samples are challenging because they are found in low quantities and have a high tendency to oxidize [19,20]. The use of many analytical techniques to determine NTs in biological samples has been developed. These techniques include chromatography with fluorescence [21,22], UV–visible spectroscopy [23], electrochemistry [24,25], high-performance liquid chromatography [26], and high-performance liquid chromatography (HPLC)–mass spectrometry (MS) [27,28,29]. Liquid chromatography–tandem mass spectrometry (LC-MS/MS) is increasingly becoming the method of choice for NT analysis due to its excellent sensitivity, specificity, and separation capability for complex samples [30,31].

The present study aimed to use *A. rosea* flower extract to treat TTX-intoxicated rats and to confirm the detoxifying effect of *A. rosea* flower by comparing the changes of NT content in brain tissues. To detect changes in the levels of nine NTs, a highly sensitive and selective LC-MS/MS approach was developed, including 5-HIAA, 5-HT, epinephrine (E), HVA, DOPAC, NE, GABA, DA, and tyramine (Tyn). Compared with the reported literature, this manuscript assays a wide variety of species that includes the less studied Tyn. The nine substances were quantified using the internal standard (IS) method, and the IS was 5-hydroxy-2-indolecarboxylic acid (5-HICA). The chemical structural formulae of the nine NTs and IS are shown in Figure 1. Behaviorally, rats after poisoning showed symptoms such as dyspnea, slow movement, and limb weakness. After treatment with *A. rosea* flower extract, most rats did not show symptoms such as limb weakness and dyspnea and were able to move normally. In terms of biological indices, after poisoning, the rats showed decreased contents of DA, NE, E, and 5-HT compared with the normal group, and the contents of GABA, Tyn, and 5-HIAA increased. After treatment with *A. rosea* flower extract, the content of these NTs was restored to normal levels. This work may provide a theoretical basis for the development of new drugs to detoxify TTX.

## 2. Results and Discussion

### 2.1. Optimization of Chromatographic Conditions

By optimizing the detection conditions of chromatography and MS, better separation and analysis were achieved. Acidic (5-HIAA, DOPAC, HVA, GABA), basic (5-HT, Tyn), and amphoteric (E, NE, DA) substances were measured; we tested water and water containing 0.1% formic acid, and 0.2% formic acid as the aqueous phase. Experimental results showed that the peak area and peak shape were the optimum when water containing 0.1% formic acid was used as the aqueous phase. Then, the ratio of water containing 0.1% formic acid (A) to methanol (B) in the mobile phase was optimized, and elution ratios of 5%B, 10%B, 15%B, and 20%B were attempted. Experimental results showed that the retention times of GABA, 5-HT, NE, E, DA, and Tyn were too short, i.e., less than 1 min, when the methanol concentration was 20%. However, the retention time of HVA was greater than 20 min and the peak shape was poor when the methanol concentration was 10% and 5%, respectively. When the methanol concentration of the mobile phase was 15%, the nine compounds were well separated within 20 min, which was the most suitable chromatographic condition. Therefore, chromatographic separation was performed using isocratic elution, and the mobile phase ratio of water containing 0.1% formic acid to methanol was 85:15.

### 2.2. Optimization of Mass Conditions

The MS/MS parameters were optimized in MRM mode by using positive and negative ion modes. MS parameters were verified and optimized for the analytes via separate infusion of each standard and IS solution into the LC-MS/MS. A mixed standard solution with a concentration of approximately 1000 ng/mL was injected into the MS system and tuned in positive and negative ionization modes. The suitable ionization mode, parent ion, two daughter ions and their corresponding cone voltage, and the collision energy of each target compound were optimized. The optimization results revealed that among the daughter ions of nine substances, the daughter ion with the asterisk had the largest response signal value, so the daughter ion was selected as the quantitative ion. Table 1 represents the optimized MRM parameters for the analytes and the IS.

### 2.3. Optimization of Sample Extract

This experiment primarily optimized the sample-preparation methods. Brain tissue is a complex biological sample in which the content of NT substances is low and the matrix is complex. Adopting a suitable pretreatment and extraction method for brain tissue was crucial. This experiment evaluated the effectiveness of six solvents for simultaneous protein precipitation and extraction, including acetonitrile, acetonitrile–water (80:20, *v*/*v*), acetonitrile–water containing 0.1% formic acid (80:20, *v*/*v*), methanol, methanol–water (80:20, *v*/*v*), and methanol–water containing 0.1% formic acid (80:20, *v*/*v*). Experimental results showed that the highest response values for each analyte were obtained when the extract solvent was acetonitrile. Therefore, acetonitrile was finally chosen as the extraction solvent.

### 2.4. Method Validation

Samples selected for validation were analyzed using the sample treatment described, and their concentrations were estimated via the IS method. The method was tested for linearity, accuracy, precision, recovery, limit of detection (LOD), limit of quantification (LOQ), and stability.

#### 2.4.1. Linearity Range and Calibration Curves

The analyte/IS peak area ratio (Y) was plotted against the analyte concentration (X) to determine the linearity of each point on the calibration curve. The standard curves for all analytes showed good linearity over a specific linear range (R^2^ ≥ 0.99), as shown in Table 2.

#### 2.4.2. Precision and Reproducibility

By using the mixed standard solutions to assess inter- and intra-day precision, the relative standard deviation (RSD) was less than 6.64%. For intra-day precision, six replicates were analyzed in one day, whereas for inter-day precision, six replicates were analyzed in three consecutive days. Reproducibility was determined by the RSD of the peak areas of compounds from the same batch of six samples. The RSDs for all samples determined were <5.07%, indicating that the method was feasible and reproducible. The results are shown in Appendix A.

#### 2.4.3. Relative Recovery and Matrix Effect

Matrix effects and relative recoveries were assessed by comparing the peak areas of standard spiked solutions of brain extracts with those of standard mixed solutions of the nine substances to be measured at the equivalent concentrations. Six samples were prepared, and a mixture of standards with the same concentration as the samples was added to samples of known concentration. The determination was repeated three times to calculate the RSD. The relative recovery and matrix effect were calculated as follows [32]:$$relative\;recovery\left(\%\right)=\frac{100\times C}{A+B}$$
$$matrix\;effect(\%)=\frac{100\times (C-A)}{B}$$
where *A* is the average peak area of the sample without spiking, *B* is the average peak area of the standard mixed solution added to the tissues, and *C* is the average peak area of rat brain tissue spiked with the same concentration as the sample.

The recoveries of the components in the samples ranged between 94.04% and 107.53%, with RSD values less than 4.21%. The RSD of the matrix effect was <4.35%, which met the methodological requirements. The results are presented in Appendix A.

#### 2.4.4. Stability

Brain tissue was placed at room temperature for 24 h, subjected to three freeze–thaw cycles at −20 °C, and placed in an autosampler at 4 °C for 48 h. The RSD of their peak area was calculated under the three conditions. The results showed that the RSD was <3.68%. This result was well within the acceptable range, as shown in Appendix A.

### 2.5. Behavioral Analysis

At lethal doses, rats quickly became lethargic and breathed ventrally. Within 10 min, the animals became immobile and their respiratory rate slowed until they stopped breathing, and death was observed within 45 min. At sublethal doses, rats initially exhibited the same clinical signs of lethargy, immobility, and abdominal breathing. Comparison between the model group and *A. rosea* flower groups revealed that most of the model group rats showed obvious signs of muscle weakness, and compared with the high- and medium-dose groups, a few rats in the low-dose group showed slight signs of muscle weakness. Results indicated that symptoms such as drowsiness, atrophy, and abdominal breathing were gradually relieved after treatment with *A. rosea* flower extract.

### 2.6. Sample Analysis

After a series of preliminary experiments, the NTs in the brain tissue of five groups of rats were quantified, and the chromatograms for multiple reaction monitoring are shown in Figure 2. The samples were diluted after the determination, and the actual sample concentrations were the measured results multiplied by two. The results are shown in Table 3. HVA and DOPAC, metabolites of NE and DA, were undetected in this study because of their low content in rat brain tissue and easy decomposition.

Table 3 shows that the contents of seven NTs changed significantly between the blank group, the model group, and the rats in the *A. rosea* flower extract administration group. Among them, the contents of 5-HT, E, DA, and NE showed a decreasing trend after poisoning. The contents of GABA, Tyn, and 5-HIAA increased after poisoning. The levels of all seven NTs gradually returned to normal levels after the administration of *A. rosea* flower extract. According to the literature, 5-HT, E, DA, and NE are NTs capable of producing pleasure. Animals with decreased levels of these four NTs show symptoms such as dyspnea, dizziness, depression, and slow heartbeat [16,33,34]. Increased levels of 5-HIAA lead to symptoms such as nerve paralysis, consistent with the symptoms of TTX intoxication. Increased GABA levels, which is an inhibitory NT, leads to lower blood pressure and promotes sleep, consistent with the behavioral changes of TTX-intoxicated rats and normal rats in the experiment [35]. Furthermore, the increase in Tyn content in the brain tissue of the poisoned rats may be related to its ability to reduce motoneuron intrinsic excitability, and the exact mechanism requires further investigation [36]. Based on the above experimental results, we proved that *A. rosea* flower extract has a certain therapeutic effect on the poisoning of TTX rats. The change in NT content may be used as a detoxification indicator.

### 2.7. Statistical Analysis

SPSS23.0 software was used to statistically analyze the NT concentrations in the brain tissue of rats in each group. The differences among the five groups of samples were examined with the post hoc comparison least significant difference method, Bonferroni t-test, and ANOVA. The results revealed that the *p* value among the blank, model, and drug administration groups were less than 0.05, which was statistically significant. The results are shown in Figure 3.

According to Figure 3, there was no significant difference between the blank group and the high-dose group in the levels of E, DA, 5-HT, and GABA, indicating that the high-dose ethyl acetate extract of *A. rosea* flower had the best therapeutic effect on these four neurotransmitters. There was no significant difference between the blank group and the medium- and high-dose groups in the levels of 5-HIAA, and the levels of the high-dose group were closer to those of the blank group, indicating a significant effect of high-dose ethyl acetate extract of *A. rosea* flower on 5-HIAA levels. However, the levels of Tyn and NE were not significantly different between the blank group and the high-, medium- and low-dose groups, indicating that low-dose ethyl acetate extract of *A. rosea* flower can reverse the levels of both neurotransmitters.

## 3. Experiment

### 3.1. Chemical and Reagents

HPLC-grade acetonitrile, methanol, and formic acid were purchased from Aladdin (Shanghai, China). Ethyl acetate, petroleum ether, and anhydrous ethanol were purchased from Sinopharm (Shanghai, China). DA, DOPAC, 5-HIAA, NE, and Tyn were purchased from Macklin (Shanghai, China). E, HVA, 5-HT, GABA, and 5-HICA (IS) were supplied by Aladdin (Shanghai, China). Unless otherwise noted, all chemical reagents were untreated and analytical-reagent grade. Dried *A. rosea* flower powder was purchased from Oriental Red Herb Company.

### 3.2. Instrumentation

The instruments used were as follows: a triple-quadrupole MS 8040 (Shimadzu Corporation, Kyoto, Japan), a UV-2450 spectrometer (Shimadzu Corporation, Kyoto, Japan), a KQ-100TDV ultrasonic water bath with temperature control (Kunshan Ultrasonic Instrument Co., Ltd., Kunshan, China), an AR224CN electronic balance (Ohaus Instrument Co., Ltd., Changzhou, China), a TG16-WS high-speed bench centrifuge (Hunan Michael Experimental Instrument Co., Ltd., Hunan, China), an SZ-93 automatic double-pure water distillatory (Shanghai YaRong Biochemical Instrument Factory, Shanghai, China), and a tissue homogenizer (Shanghai Jingxin Industrial Development Co., Ltd., Shanghai, China).

### 3.3. Preparation of A. rosea Flower Extract

About 25 g of dried powder of *A. rosea* flower and 250 mL of anhydrous ethanol were placed in a 500 mL round-bottom flask, extracted under reflux for 2 h, and filtered. The residue was extracted again by repeating the above method. The filtrates were combined and concentrated with rotary evaporator. The obtained solid extract was placed in a vacuum drying oven at 50 °C for 5 h and extracted with ethyl acetate three times. The solvent was rotary-evaporated to obtain the ointment, which was then lyophilized into powder. The lyophilized powder of *A. rosea* flower extract was weighed to 3200 mg and fixed in a 100 mL volumetric flask with saline to prepare a 32.0 mg/mL stock solution. The stock solution was stored in the refrigerator at 4 °C and subsequently used for dilution to prepare the solution at the concentration required for treatment in rats [37,38,39]. The chemical composition of the ethyl acetate extract of *A. rosea* flowers was studied. A total of 28 chemical components were identified via LC-Q/TOF-MS. For information on these compounds, see Appendix A.

### 3.4. LC-MS/MS Conditions

A Shimadzu triple-quadrupole MS was interfaced with a Shimadzu HPLC system by using an electrospray ion source. MS detection was performed in positive-ion mode through multiple reaction monitoring (MRM). Analytes were separated on a Welch AQ-C18 column (2.1 mm × 100 mm, 3 μm) operated at 35 °C. The mobile phase, comprising 0.1% formic acid in water (aqueous phase A) and methanol (organic phase B), was delivered at a flow rate of 0.2 mL/min, and the elution concentration was 15% B. The injection volume was 5 μL, and the total time taken for the chromatographic run was 20 min per sample.

### 3.5. Standard Solution

The standard stock solutions (1 mg/mL) of the nine analytes, namely, NE, E, 5-HT, DOPAC, HVA, 5-HIAA, Tyn, and 5-HICA (IS), were prepared by dissolving the appropriate amount of the standard substance in acetonitrile. The standard stock solutions of DA and GABA were prepared in acetonitrile–0.1% formic acid water solution (80:20, *v*/*v*). The stock solutions of the analytes were further diluted with methanol to prepare working standard solutions at the desired concentrations.

### 3.6. Animals

Fifty male Wistar rats weighing 120–150 g were selected for formal experiments and randomly divided into five groups: blank, model, and *A. rosea* flower therapy groups (high, medium, and low doses). Each group had 10 rats. Prior to the experiments, the animals were given adequate food and water and were housed in an approved facility that maintained a 12 h light/dark cycle. All animals were purchased from Beijing HFK Bioscience Co., Ltd. (SCXK 2019-0008, Beijing, China), inspected by the Institute of Medical Laboratory Animals, Chinese Medical Sciences and approved by the Ethics Committee of Fujian Medical University for the project under ethics number IACUC FJMU 2022-0492.

### 3.7. Establishment of the Model

Based on the available literature, a Wistar rat TTX poisoning modeling protocol was designed for a multiple-dose intramuscular toxicity pretest with seven dosing groups of 7, 8, 9, 10, 11, 12, and 13 μg/kg, respectively, with three male rats in each group. Rats that did not die after 1 h could be used as the initial model for TTX poisoning. After three modeling experiments, the modeling dose was determined to be 11 μg/kg [40].

### 3.8. Experimental Design and Treatment

In pre-experiments, ethyl acetate *A. rosea* flower extract was found to be the most effective at treating TTX. Animals were acclimatized to laboratory conditions before experimentation. After poisoning the rats on the first day, drug delivery was continued with the ethyl acetate extract of *A. rosea* flower for 4 days, once a day. The protocol details are shown in Table 4. After completion of the experiment, brain tissue was removed and stored in a −20 °C refrigerator until analysis.

### 3.9. Sample Preparation

A total of 0.1000 g of rat brain tissue was added to 0.50 mL of acetonitrile and homogenized with a tissue homogenizer for 30 s. After vortex mixing for 30 s, the homogenate of the tissues was centrifuged at 12,000 rpm for 10 min. The supernatant was obtained and the residue was re-extracted using the same method. Then, 0.5 mL of the combined supernatant was precisely pipetted and an equal volume of acetonitrile added to dilute it and reduce the effect of the sample matrix. Diluted samples were filtered and analyzed using LC-MS/MS. The sample extract and detection process are shown in Figure 4.

## 4. Conclusions

A highly selective, sensitive, and rapid analytical method of UPLC-MS/MS was developed for the determination of nine NTs in rat brain tissues with good precision and accuracy through a series of methodological validation. The experimental results showed that the levels of all seven NTs in the brain tissues of rats changed significantly after poisoning and gradually returned to normal levels after the administration of *A. rosea* flower extract. The results proved that TTX poisoning affected the content of the endogenous substance Tyn, and also confirmed the detoxification effect of *A. rosea* flower on TTX. This study provided experimental methods and a theoretical basis for the development of new TTX detoxification drugs in clinical practice. However, further analysis of the extract fractions is needed to identify the active components of *A. rosea* flower that play a detoxifying function.

## Figures and Tables

**Figure 1 molecules-28-04158-f001:**
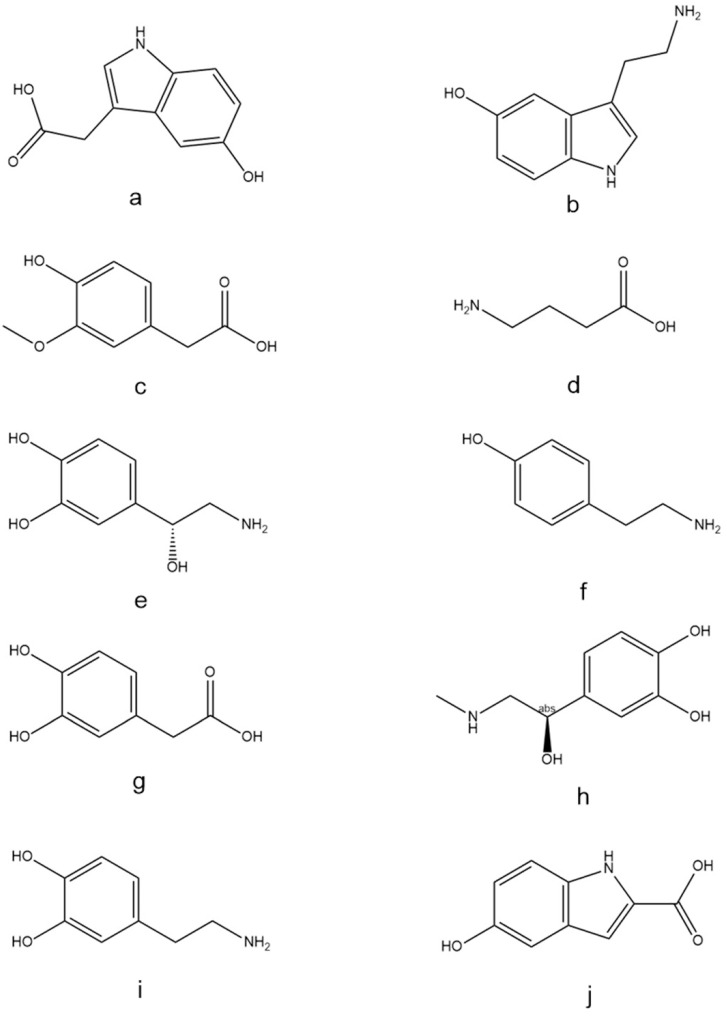
Chemical structures of the targeted NTs: (**a**) 5-HIAA, (**b**) 5-HT, (**c**) HVA, (**d**) GABA, (**e**) NE, (**f**) Tyn, (**g**) DOPAC, (**h**) E, (**i**) DA, and (**j**) 5-HICA (IS).

**Figure 2 molecules-28-04158-f002:**
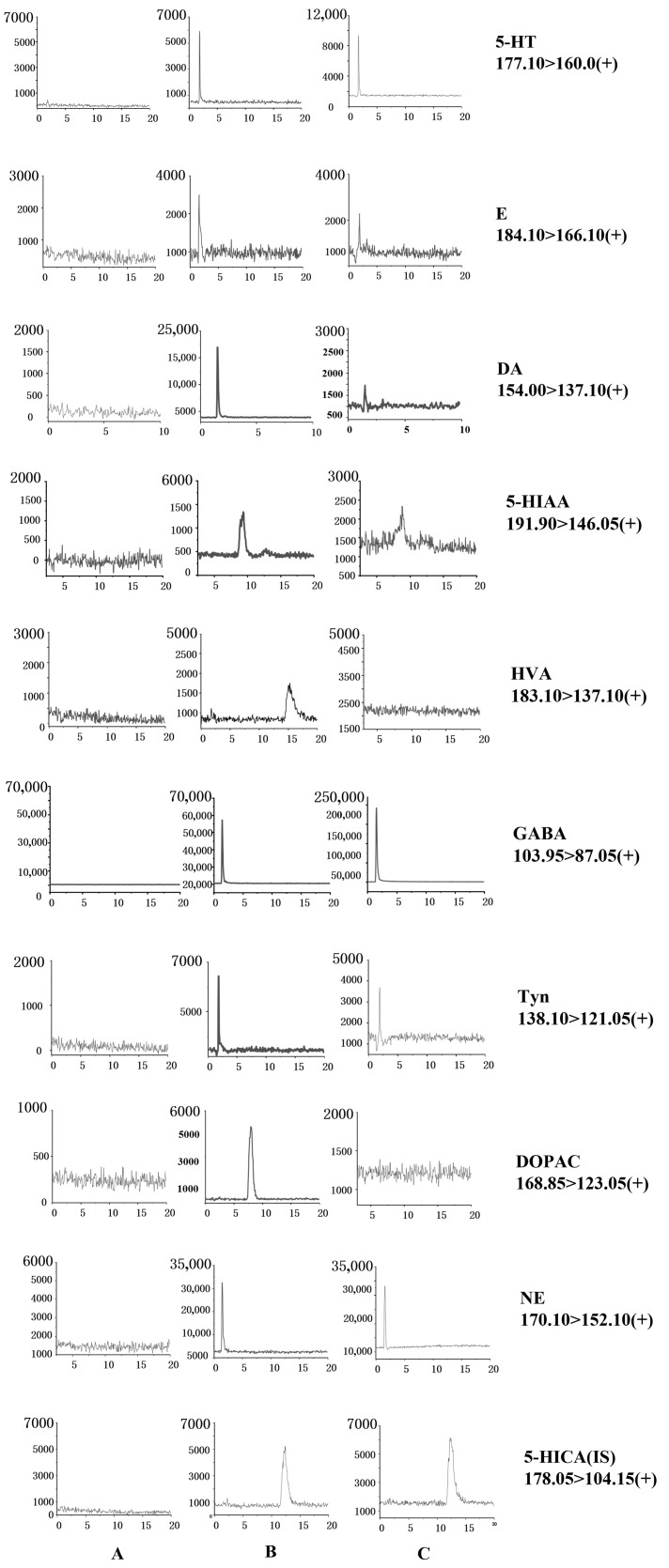
Chromatograms of nine analytes and internal standards: (**A**) blank solvents; (**B**) standard substance; and (**C**) samples.

**Figure 3 molecules-28-04158-f003:**
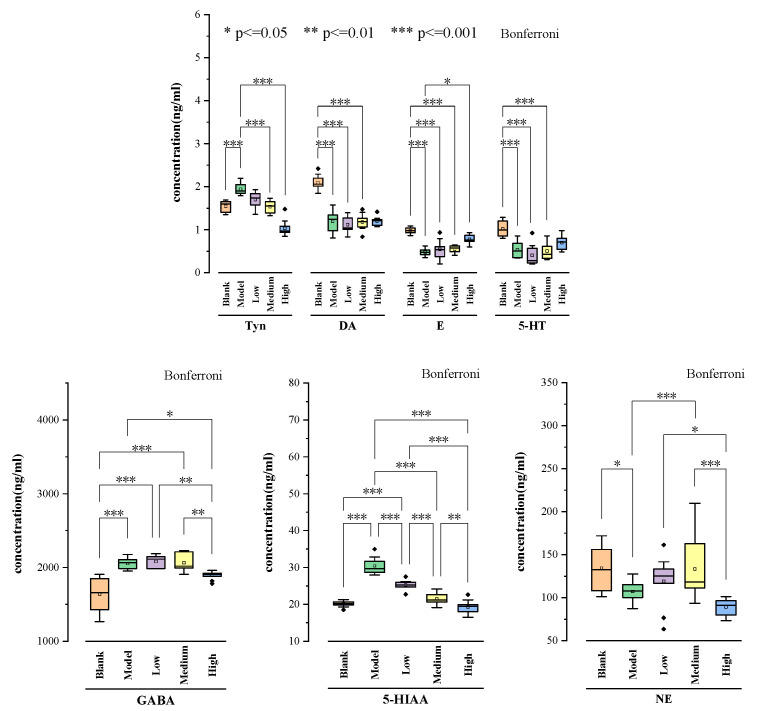
Statistical analysis of seven neurotransmitters in rat brain tissue among the blank, model, and drug administration groups.

**Figure 4 molecules-28-04158-f004:**
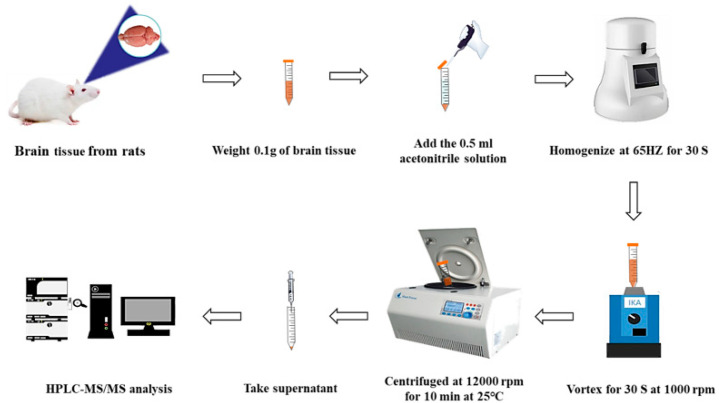
Extract and detection workflows of neurotransmitters in rat brain tissues.

**Table 1 molecules-28-04158-t001:** MS parameters of nine analytes and IS.

Analytes	Q1 (*m*/*z*)	Q3 (*m*/*z*)	CE (eV)	t_R_ (Min)
HVA	183.10	137.10 */122.03	−14.9	15.20
5-HIAA	191.90	146.05 */72.10	−13.0	9.25
DOPAC	168.85	123.05 */123.99	−12.0	6.30
GABA	103.95	87.05 */68.90	−14.0	1.41
5-HT	177.10	160.0 */132.08	−10.0	1.69
NE	170.10	152.10 */44.04	−10.0	1.52
DA	154.00	137.10 */91.05	−15.0	1.42
E	184.10	166.10 */135.04	−15.0	1.40
Tyn	138.10	121.05 */122.05	−10.0	1.60
5-HICA(IS)	178.05	104.15 */160.05	−12.0	12.40

* Transition used for quantification.

**Table 2 molecules-28-04158-t002:** Linear ranges, detection limits, and quantitation limits for compounds.

Analytes	Calibration Curves	R^2^	Linear Range (ng/mL)	LOD (ng/mL)	LOQ (ng/mL)
HVA	y = 0.0125x − 0.2686	0.9942	50–800	15	45
5-HIAA	y = 0.4158x − 0.0047	0.9968	4–40	1	3
DOPAC	y = 0.3578x − 0.8501	0.9941	5–50	2.5	5
GABA	y = 0.1554x − 2.1174	0.9994	150–3000	30	100
5-HT	y = 5.2507x + 0.3338	0.9997	0.3–3	0.01	0.03
NE	y = 0.3128x + 1.6218	0.9997	4.5–90	1	3
DA	y = 0.7623x + 0.0024	0.9982	0.75–15	0.2	0.5
E	y = 2.5693x + 0.7171	0.9991	0.2–5	0.02	0.05
Tyn	y = 3.3475x + 0.4305	0.9967	0.2–5	0.03	0.1

**Table 3 molecules-28-04158-t003:** Content of nine analytes in the brain tissue of five groups of rats (mean ± SD) (*n* = 10).

Analytes	Blank Group (ng/g)	Model Group (ng/g)	Low-Dose Group (ng/g)	Medium-Dose Group (ng/g)	High-Dose Group (ng/g)
HVA	-	-	-	-	-
5-HIAA	402.51 ± 0.74	609.07 ± 2.10	505.22 ± 1.17	429.34 ± 1.38	386.08 ± 1.68
DOPAC	-	-	-	-	-
GABA	32,767.73 ± 228.54	41,078.20 ± 69.23	41,590.95 ± 79.21	41,308.83 ± 118.17	37,867.59 ± 51.88
5-HT	20.38 ± 0.17	10.58 ± 0.17	8.16 ± 0.23	10.12 ± 0.20	14.03 ± 0.16
NE	2686.34 ± 23.70	2144.23 ± 11.22	2379.54 ± 27.44	2665.02 ± 34.79	1785.17 ± 8.69
E	19.53 ± 0.07	9.65 ± 0.08	10.90 ± 0.19	11.07 ± 0.07	15.58 ± 0.09
DA	41.91 ± 0.16	23.95 ± 0.23	22.31 ± 0.18	23.56 ± 0.18	24.00 ± 0.09
Tyn	30.85 ± 0.13	38.84 ± 0.12	34.01 ± 0.17	30.71 ± 0.14	20.69 ± 0.18

**Table 4 molecules-28-04158-t004:** Experimental protocol for five groups of rats in 4 days.

Groups	Types and Modes of Poisoning (11 μg/kg)	Type and Mode of Drug Delivery (4.0 g/kg)
Blank group	Physiological saline (i.m)	Physiological saline (i.g)
Model group	TTX (i.m)	Physiological saline (i.g)
Low-dose group	TTX (i.m)	8 mg/mL *Althaea rosea* flower ethyl acetate extract (i.g)
Medium-dose group	TTX (i.m)	16 mg/mL *Althaea rosea* flower ethyl acetate extract (i.g)
High-dose group	TTX (i.m)	32 mg/mL *Althaea rosea* flower ethyl acetate extract (i.g)

## Data Availability

The original contributions present in the study are included in the article/Appendix A. Further inquiries can be directed to the corresponding authors.

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
