# Peer review of "Determination of Multiple Neurotransmitters through LC-MS/MS to Confirm the Therapeutic Effects of Althaea rosea Flower on TTX-Intoxicated Rats"

_molecules, 2023, doi:10.3390/molecules28104158_

Round 1
Reviewer 1 Report
Treatment for tetrodotoxin poisoning mainly involves respiratory support and supportive care until the toxin is excreted in urine . Currently, there is no known antidote for tetrodotoxin, but some drugs such as Neostigmine have been used to treat acute respiratory failure caused by the poisoning . Additionally, prophylaxis and treatment with a monoclonal antibody have shown promise in animal studies . Other measures that can be taken include gastric lavage and/or oral ingestion of activated charcoal , IV hydration, hemodialysis, and standard life-support measures. It is important to note, however, that the severity of the poisoning can vary widely and survival rates can be low. If someone has ingested tetrodotoxin, it is critical to seek immediate medical attention. Nevertheless, the authors presented a mediocre antidote tested on mild developed animal intoxicated pharmacological model. Mild amelioration has been noticed in the reported figures which lack the statistical significance testing using t-test or ANOVA. Catastrophically, the chemical content of the displayed ethyl acetate extract of Althaea rosea flower were not studied. Unfortunately, without the metabolomic profile of the extract and lack of statistical significance testing and poor design of the experiment I will be going to reject the current submission unless substantial changes are considered.
Reviewer 2 Report
This research article introduces a study on the potential of the ethyl acetate fraction extract of A. rosea flower in treating TTX poisoning in rats by LC-MS/MS detection. The study revealed that the levels of all seven neurotransmitters returned to normal after the administration of ethyl acetate extract of A. rosea flower, indicating its therapeutic effect on TTX poisoning and contributing to the field of pharmacology. However, there is a need for a major revision of the experimental part (calibration curves) to improve the clarity and presentation of the data. Further details regarding the necessary changes are outlined below:
1. P.2, “The accurate determination of NT content changes in biological samples before and after TTX poisoning is challenging due to low NT content, chemical instability, and severe matrix effects in biological samples[30]” is redundant since “…biological samples are challenging because they are low quantities and have a high tendency to oxidize” is at the beginning of this paragraph?
2. P.2, “HVA, DOPAC, DA, GABA, DA”, two “DA” and please correct it.
3. P.3, change “these NTs’ content was” to “the content of these NTs was”.
4. P.6, “Acidic and basic substances were measured”, please specify “Acidic and basic substances”.
5. P.7, change “precipitation and extract” to “…extraction”.
6. P.7 Table 3, “R2≥0.99” is not strong enough to state a good linearity. R2≥0.999 would be much convincing.
7. P.7, please remove this caption, “Table S1. Summary of intra- and inter-batch precision and repeatability data in rat brain homogenate”, if this table is not shown in the main text.
8. P.8, please remove two captions, “Table S2…” and “Table S3…”, if these two tables are not shown in the main text.
Round 2
Reviewer 1 Report
Thanks for the reply and the amended version which is way improved. However, the gradient in the supplementary information2 Table 1 is incorrect. I believe at 2 min A% should be 95 and B% Should be 5%. Please mention the acquisition mode which I believe is AutoMSMS. Based on the selected mobile phases (Formic ACN/Water) it is advised not to include less possible adducts like acetate and ammonium which may introduce false positives. Therefore, please kindly revisit the annotation of compound 24 and 29 which were annotated as NH4 adducts although no ammonium buffer was used. Overall, I am pretty happy with the amended figures and where the stats added confidence and significance to the overall study. Figure 3 label amendment to the abundance of the measured neurotransmitter will be more accurate and ament the Y axis to concentration (ng/ml). Also, the no significant differences between the treated groups and control group are to be listed (crucial) and, in the figures, and to be discussed as it will be more relevant to the effect compared to the statistically significant difference from the model. delete Figure 4, it is a bar chart presentation of figure3 boxplots. All the best.
Reviewer 2 Report
Thank you for addressing these questions.
Author Response
Thank you for reviewing the manuscript!